# Tutorials for Children by Children: Design and Evaluation of a Children's Tutorial Authoring Tool for Digital Art

Ananta Chowdhury, Andrea Bunt

Department of Computer Science,

University of Manitoba

**ABSTRACT**

Digital art tools allow children to express their creativity and can help them develop important skills. There are numerous software tutorials available to help teach and inspire digital art enthusiasts, however, most are authored for and by adults. Given that children are increasingly contributing online digital content, in this paper, we investigate a tutorial authoring design concept where children can capture their drawings and information on their process, with the long-term objective of allowing children to share both their creativity and their workflows with other children. Through participatory design sessions, prototyping, and an evaluation, we explore children's attitudes towards the creation of digital art tutorials, focusing on their perceived incentives to author such tutorials and how they feel about the concept of sharing their tutorials with other children. We also elicit reactions towards specific design elements. Our findings suggest important considerations for tools designed to motivate and support children's creation of digital art tutorials.

**Keywords**: Digital art, Drawing, Tutorial authoring system, Sharing workflows, Child-computer interaction, Peer-based learning.

**Index Terms**: H.5.2 [Information Interfaces and Presentation (e.g., HCI)]: User Interfaces; H.5.m [Information Interfaces and Presentation (e.g., HCI)]: Miscellaneous—User studies, Participatory design

## 1 INTRODUCTION

Art is a common way for children to express themselves. Engaging in art and creativity is considered a productive use of children's time, by promoting social, emotional, motor, and cognitive development [4,44], providing a sense of accomplishment, and boosting self-esteem [37]. Digital art tools allow for new effects, many of which are not possible with physical drawing tools. To inspire children to create digital art and connect with other art enthusiast peers, there are several digital art platforms that provide child-centric areas for children to share their creations [69,70].

While sharing digital art, many adult creators share not only their end products but also step-based instructions on how they used a particular feature-rich software tool to create them. In doing so, tutorial authors can both showcase their skills and creativity, and help others learn how to use feature-rich software tools to produce similar effects [38,42,52]. With these advantages in mind, prior

---

\* chowdhu1@myumanitoba.ca, andrea.bunt@umanitoba.ca

research has contributed several tutorial systems and authoring tools to support this process [14,19,21,27,40].

Despite the potential advantages of creating and sharing digital art tutorials and the fact that children are already actively sharing digital art online, research on tutorial creation tools has generally focused on adults. In addition to showcasing skills and creativity, generating tutorials for peers would provide children with the opportunity to take on the role of a tutor, which has been shown to help children learn to think from others' perspectives, grow their sense of responsibility [2], and foster self-acceptance [1]. Along with developing useful skills such as planning and communicating, enacting the role of a teacher while creating digital tutorials can provide children with a sense of ownership, and purpose [47].

In this research, we explore children's attitudes towards creating digital art tutorials for their peers and how a tutorial authoring system might support them in doing so. Our investigation centres around the following research questions: 1) Are children interested in authoring drawing tutorials for other children while creating digital art? 2) What do they see as potential benefits or incentives? 3) How might a semi-automated tool support children in creating tutorials? 4) How do children use a semi-automated tutorial authoring system to communicate their digital art workflows?

To address our research questions, we used prototyping as a means of inquiry to elicit reactions and input from our target population. We first conducted a formative study with eight children (ages 6-11) using paper prototyping to evoke responses towards an initial tutorial authoring concept and to refine individual design elements. In a second study with 16 additional children (ages 7-11), we used a higher-fidelity prototype to further probe on attitudes towards creating tutorials, as well as how children might use such a tool. Findings from our study suggested that many children are interested in creating tutorials, with perceived incentives ranging from altruism, to showcasing drawing skills, to documenting their workflows for their own recollection later. Children used the higher-fidelity prototype to generate a range of creative tutorials, indicating the potential of a semi-automated tutorial authoring system to support children's tutorial creation while producing digital art. In comparison to a sample of adult-authored tutorials, the tutorials created by our participants had a similar number of steps, but focused less on structure and more on being creative with their drawings. Our findings also highlight considerations for child-centric authoring tools, such as the importance of balancing tutorial creation with drawing and providing scaffolding to help children annotate their tutorials.

The paper's contributions are as follows: 1) We present findings from two studies that illustrate children's attitudes and approaches to creating digital art workflows for their peers. 2) Through an iterative design and evaluation process with children, we provide insight into how an authoring system can support children in creating digital art tutorials.

## 2 RELATED WORK

In this section, we first discuss prior research on tutorial systems and tutorial authoring tools. We then briefly discuss previous research demonstrating the potential for children to create tutorials for their peers. Finally, we turn to research on children creating different kinds of online digital content.

### 2.1 Tutorial Systems and Tutorial Authoring Tools

Digital art is often created using complex software, which has been the focus of a large body of work on designing tutorials and other help systems to support their use [35]. For example, several studies have concentrated on generating image-based tutorials by capturing and visualizing users' operation history of using an application [27,32,46]. There are also systems that automatically generate tutorials containing both the workflow histories and videos of the operations [14,28]. Our work is informed by these prior authoring systems; however, whereas the above work has focused on adults, we specifically focus on a system to help children create tutorials, involving them in the design process.

Also relevant to our work are systems that assist users with digital drawing, for example, by providing guidance on how to attain certain effects or drawing elements [19,21,33,40]. Other work has focused on assisting children in applying tutorials, by helping them locate relevant elements in the target software [30]. In our work, we focus on tools to support children in documenting their drawings and processes. As such, we see our work as being complementary to but distinct from this prior work on helping users (including children) achieve greater drawing success.

Although we are not aware of prior work examining tutorial authoring tools for children, there are online platforms for sharing digital art and tutorials with a degree of child focus. For example, DragoArt [71] and DrawingNow [72] list some drawing tutorials targeted at children, however, the vast majority are authored by adults or staff illustrators. Our work focuses on involving children in the design process and on eliciting their reactions to creating tutorials for other children.

### 2.2 Children Tutoring Their Peers

Our work builds on previous research showing that children can author academic digital tutorials for their peers [47,48] and other teaching-oriented resources, such as educational games to teach other children [34]. Art tutorials differ from those investigated above in that they have the potential to focus more on creativity and inspiration generation than on teaching specific topics. For example, a child-generated math tutorial is created to help peers understand and review a mathematical concept [48], whereas a drawing tutorial might serve to inspire artistic creativity in others. Recent research showed the potential of a music learning app, where children recorded their piano pieces and tutorials on different practice strategies and shared those in an online space to encourage and help their peers to learn how to play piano [11]. This research shared similar motivations as ours – inspiring creativity and supporting peer-based learning while enabling children to showcase their artistic competency.

Authoring content for other children can help children develop a variety of skills. For example, researchers have investigated the design of game-authoring tools for children [24,68] since game creation has the potential to develop narrative skills, improve critical thinking, computer and media literacy, and boost self-esteem [3,24,34]. Collaborative storytelling authoring tools [57,64] improve children's communication skills and writing abilities [57]. Motivated by these benefits, we explore children's attitudes towards a tutorial authoring system that allows them to create digital art tutorials for other children.

### 2.3 Children's Creation of Different Creative Digital Content

Our work is also inspired by prior work showing that children are interested in and capable of generating creative digital content with the purpose of sharing this content with others. For example, online programming environments like Scratch [54] provide children with the opportunity to create their own interactive digital content, share ideas, collaborate, and communicate with like-minded peers [12,16,56]. Interactive digital storytelling platforms also allow children to practice creativity by generating imaginative stories and collaborating with others [5,8,26,31,41]. In another vein, online user-generated video-sharing communities like YouTube are becoming increasingly popular among children as a stage to exhibit their skills [67] and engage actively with their audience [43]. Digital art creation is but another way for children to express their creativity. Hence, to support children's creation of digital art, researchers have focused on children's cooperative drawing approach [58] and proposed tools to promote collaboration among peers [7,20]. Findings from these studies suggest that appropriately designed tools to create digital content can provide children with the opportunity to express themselves [6,43,67], showcase their innovativeness [5,12,16,54], and also inspire others to participate and collaborate [8,26,31,56]. These findings motivated us to investigate how children would approach a tutorial authoring system where they can create digital art tutorials as guidelines for their peers while showcasing their digital art skills.

## 3 AUTHORING AND SHARING WORKFLOWS: GENERAL APPROACH

To generate insight into how children respond to the idea of documenting and sharing their digital art workflows, we used prototyping as a means of inquiry. Based on previous research showing the value of low-fidelity (lo-fi) prototyping in designing child-oriented applications [23,45,55,60,62], we started with a paper prototype, which we used to elicit initial reactions in a formative study. We then used insights from this formative study to develop a higher-fidelity prototype that we used to conduct a more detailed evaluation.

After exploring prior work on tutorial authoring [14,19,27,32,33,35,42,46], we used sketching to explore features that could facilitate a child's tutorial creation process. For example, we considered automatically capturing screenshots or videos of drawing steps (i.e., after each tool use or drawing modification), enabling children to capture their own steps while drawing, and allowing children to create tutorial steps later from a recorded video of their drawing process. In comparing alternatives, our goal was to keep the tutorial creation process simple, to provide some autonomy, and to avoid detracting too much from the fun of drawing.

After our sketching process and review of prior work, we settled on an initial design direction that involves allowing children to capture information on their workflows while they are drawing. Based on prior work showing that most tutorials follow a step-based nature [27,42], our tutorial authoring approach assists the child in recording and documenting individual steps of their drawing. In our approach, the child decides when they are ready to save a step, with the prototype capturing the image and information regarding the tools used during that step. To communicate information about their drawing to others, we let children provide comments or tips associated with their steps since prior work suggests that instructions including a combination of images and text are more useful than those that rely on either images or text in isolation [27]. Finally, we wanted to include a review component, where the child could potentially modify their tutorial before saving it and/or sharing it. Our current design approach does not include

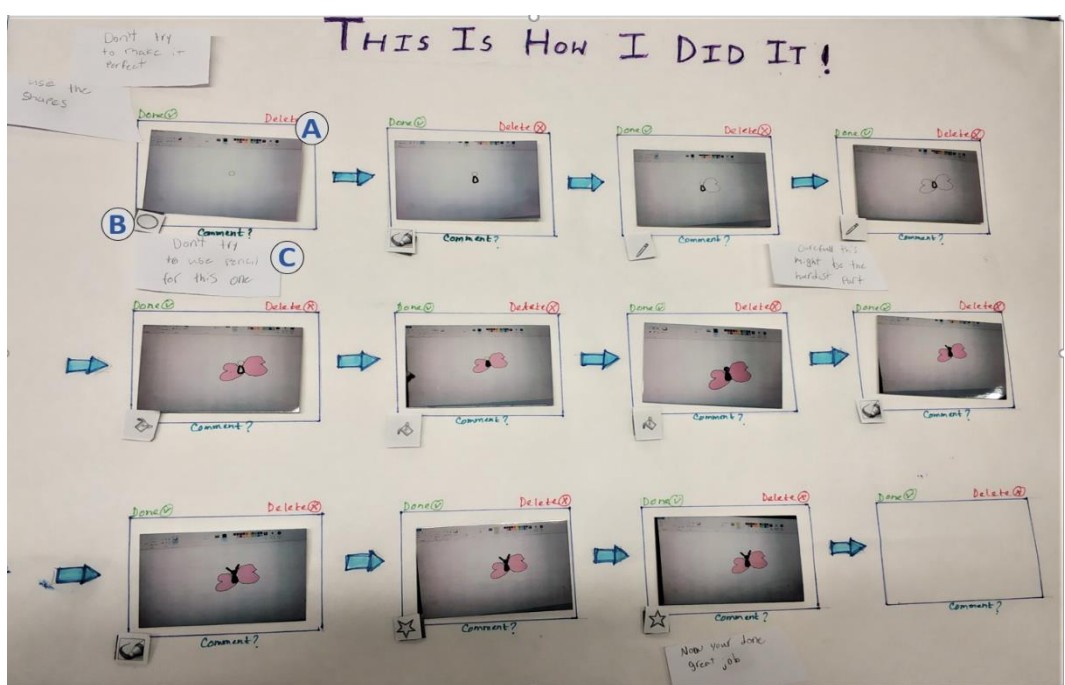

Figure 1. Low-fidelity Prototype. The workflow was generated by a 9-yr-old girl participant in our formative study; The prototype depicts a series of steps to the drawing as defined by the participant. (A) Each box contains a picture of the image that the participant generated using the drawing program for that step. (B) The icon at the bottom left corner of each box indicates which tool was used during that step. (C) The participant also provided tips or comments on pieces of paper and attached them with her captured steps.

video demonstrations. We made this design decision based on previous research indicating that navigating video or animations can be complex and time-consuming [27,46], however, adding video elements could be explored in future research.

Our initial target audience for this approach was children who were 6-11 years old. We targeted this range to cover children who can think logically and make independent decisions (ages 6-10) [18] and who can reason inductively and think from others' perspectives (ages 7-11) [51]. We refined our target age to 7-11, based on observations from our formative study.

### 3.1 Low-fidelity Prototype

To explore the general authoring approach described above with children, we created a low-fidelity prototype. The lo-fi prototype is a paper-based template for a tutorial authoring and display system that has slots for each step in the tutorial (Figure 1). The steps are determined by the child while they are drawing: when they feel that they have reached a step in their workflow, an image of that step is added in the next available slot. Each step also includes sticky notes for the tools used and any comments the child provides. Figure 1 illustrates an example of a complete workflow created by a child with our prototype.

A challenge that we faced while paper prototyping was simulating the tools (e.g., colour effects, undo/redo, copy and paste) of a digital drawing application on paper in a way that would be engaging for children. So, instead of drawing on paper, we decided to let children draw using Microsoft Paint, which meant that we needed a way to transfer different states of their drawing to the paper prototype. We used a camera and a Polaroid printer for this purpose to capture the image on the screen and quickly print a photo to attach to the paper prototype. This enabled a child to work with the compelling drawing tools, while still retaining the advantages of paper prototyping for eliciting design feedback.

## 4  FORMATIVE STUDY

We used our lo-fi prototype in a formative study to elicit initial reactions from children on the idea of sharing digital art along with how they made it. To design appropriate child-oriented technology, prior research has recommended involving children in the design process by adopting and extending participatory design methods [10,11,17,29,41,59]. Inspired by this body of research, we also conducted participatory design sessions with the children to refine our authoring system concept. Throughout these participatory design sessions, we encouraged the children to share their ideas while interacting with our low-fidelity prototype.

### 4.1  Participants

We recruited 8 participants (5 girls, 3 boys) locally, who were 6-11 years old with previous experience in digital drawing through snowball sampling [25] and by placing advertisements throughout our university campus, situated in Canada. We also asked for a parent to participate in the study so that we could interview them regarding any concerns they might have. After receiving written consent from the parents and verbal assent from the children, we initiated the study sessions. We informed the child that they could withdraw from the study at any time. In appreciation of their time and participation, the children received a small toy of their choice, and the parents received $15 in cash. The study was approved by our research ethics board.

Of the eight participants, the two six-year-olds had difficulty grasping the idea of capturing the workflows of their digital art. The remaining six older children (i.e., 7-11) seemed to understand the concept and were therefore in a better position to provide concrete feedback. We report on findings from these six children in the 7-11 age range. We also used these observations to adjust the target age range for our second study.

### 4.2 Study Tasks, Procedure, and Data Collection

The study was conducted in a research laboratory (pre-COVID) with one participant at a time. As per our institutionally approved protocol, the child's parent was also present during the entire session. The main tasks in these sessions involved the child creating a digital drawing while a researcher helped them capture their drawing steps. Using the lo-fi prototype, the child then worked with the researcher to craft a tutorial.

To help the children understand the context of the use of our prototype, we started the study session by demonstrating a storyboard prototype (see the Supplementary Material). The storyboard introduced the idea of creating a tutorial by depicting a scenario, where a child creates a tutorial and shares it with her friends to illustrate her drawing workflow. Next, we asked the child a few interview questions about their thoughts on sharing their drawings and workflows, seeing others' drawings, and following others' workflows. We then showed the child a PowerPoint prototype to demonstrate what capturing steps of their drawing might look like, before asking the child to draw using MS Paint.

After we had introduced the tutorial concept, we asked the child to create a drawing to include in a tutorial. We encouraged the child to tell us when they were ready to create a step, at which point we took a picture of their screen with our camera.

When the child was done with the drawing, we printed the captured photos. Then, the child and the researcher started pasting the photos on the prototype. We had a set of small sticky notes with icons of different drawing tools that the child could attach under each step. They could also write tips and comments on pieces of paper and attach those to the steps. We asked them about what they liked and did not like about the prototype, what they would want to change, and what other information they thought might be useful for another child wanting to follow their tutorial. During this process, we encouraged the participant to draw and sketch on the paper prototype to demonstrate their design ideas.

We concluded our study session by interviewing the child's parent about any concerns they might have regarding children's sharing of digital art and workflows. Each session lasted approximately one hour.

We collected data from our participatory design sessions, the semi-structured interviews with the children, and the short semi-structured interviews with their parents. We video-recorded the participatory design sessions and audio-recorded the interview sessions, which we transcribed and analyzed using open coding to identify participants' views (both positive and negative) towards our tutorial authoring approach and specific design ideas. While qualitative analysis should not necessarily yield counts, we felt that we could see clear enough boundaries in participant views to include counts in our reporting. We do so to give a sense of how prevalent certain sentiments were in our data.

### 4.3 Findings

#### 4.3.1 Feedback from the Children

Upon asking whether they would like to share their drawings with others, most of the participants (5/6 participants) expressed enthusiasm for the idea of sharing drawings and workflows to showcase their drawing skills and also to help others attempt to recreate their drawings.

*Then someone can do that too and then they'll be happy too.*
*— P3 (7-yr girl)*

Only one participant was hesitant to share his drawing because he felt that it was not good enough, suggesting a lack of confidence.

All six participants were interested to see other children's drawings. They found this concept entertaining and thought it would help them generate ideas. All participants also expressed interest in seeing the workflows behind these drawings. They felt it would help them to recreate a particular drawing they liked.

*Once my friend Danny, she drew a really cool thing like a girl, and I was like how did you do that?! I would like to try that. – P6 (9-yr girl)*

From our participatory design sessions, we observed that all six children who were 7-11 years old understood what steps are in a workflow. All six liked the sequential way of displaying the steps as shown in Figure 1. They also found the icons of the tools associated with each step helpful. They believed the display of the workflows was simple and intuitive for other children to understand the drawing process.

*I like this because if you are reading a book, you'll go like this. – P3 (7-yr girl)*

All participants created multiple steps to illustrate their drawing process. They did not hesitate to let us know to capture a photo of the drawing to make it a step. However, sometimes, when concentrating intently on their drawing, a few children (3/6) forgot to capture some of their steps. To tackle this, one participant suggested showing reminders. Participants did not, however, want the system to capture steps without their permission – they wanted to remain in control.

In terms of annotating their workflows, while most children were reluctant to write comments at the beginning, everyone attached at least one comment. Examples of their comments included: "Don't try to use pencil for this one", "Careful, this might be the hardest part!", "Now you're done. Great job!". One participant mentioned that having the option to write comments while saving the steps would be more beneficial as they might think of a comment while drawing a particular step and forget about it later.

#### 4.3.2 Feedback from Parents

In general, the parents were not concerned about their children sharing their drawings online (6/6). Their main concern was that appropriate parental controls be in place to control what children are sharing and with whom (3/6). A concern more specific to this sharing domain was children sharing art tutorials might affect their creativity negatively if they always try to follow others' instructions (2/6). On balance, the parents tended to feel that the opportunity to learn to draw from other children would have positive effects (4/6):

*Sometimes learning to do something somebody else's way can kind of encourage you and give you ideas for how to do something your way. I don't think it'll stifle her creativity as long as she has time and space to do her own things too.*

To summarize, in response to our first research question of how children would feel about the idea of creating tutorials to share with other children, our formative study provides preliminary insight that our participants were generally positive about the idea. We did see some hesitance that might be attributed to lower confidence, however, warranting further study with a larger sample. In terms of our specific design approach, which borrows elements from adult-oriented tutorial authoring systems (e.g., sequentially displayed steps, commenting, etc.), our participants were generally comfortable with the main interaction style and provided feedback on how to further improve it to meet their needs (e.g., step capture reminders and more flexible commenting). The parents responded positively to the idea of their child sharing their drawings with others, provided proper parental controls were in place.

## 5 DEVELOPING A HIGHER-FIDELITY PROTOTYPE

In the next phase of our research, we converted our lo-fi prototype

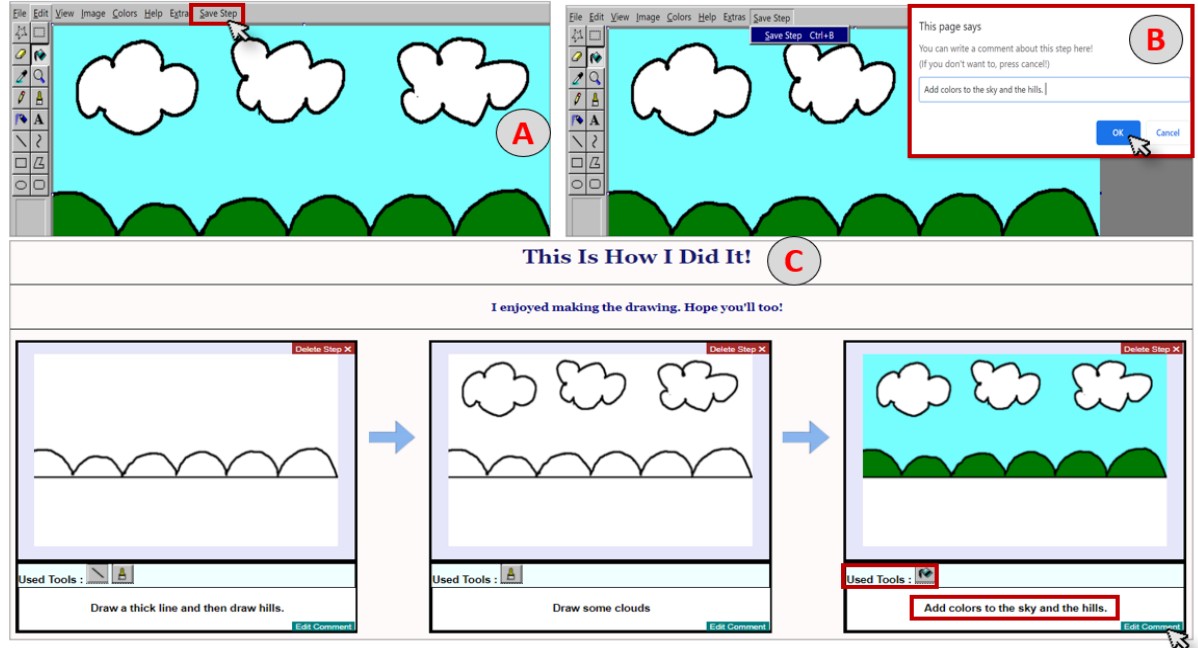

Figure 2: Higher-Fidelity Prototype: (A) A "Save Step" feature has been added to JS Paint. When clicked, the prototype captures the progress of the drawing along with tools used for that step; (B) The child can optionally choose to provide a comment with the step; (C) Upon completion of the drawing, the prototype displays the captured steps sequentially along with the associated comments and used tools. Children can edit both the comments and the tools.

into a higher-fidelity one by incorporating children's feedback from the formative study. In creating the higher-fidelity prototype, our goal was to use it as a means of further inquiry [63]. Specifically, we wanted to use this higher-fidelity prototype to gain more detailed insights into how children might respond to our tutorial authoring approach. To facilitate our prototype development, we used a mix of automated capture and Wizard-of-Oz techniques.

Our higher-fidelity prototype (Figure 2) allows a child to generate a tutorial while drawing digital art by enabling them to self-capture, annotate, and edit their drawing steps. Our prototype currently works with JS Paint [49] (Figure 2A), an open-source drawing program. While using this program to draw something, when the child chooses to capture a step by clicking the "Save Step" feature (Figure 2A), the prototype automatically records the current state of the drawing as well as the tools used as part of that step. The prototype also allows the child to add a comment when saving a step (Figure 2B). This design decision was based on the feedback from our formative study that some children preferred to write comments while working on the drawing to avoid forgetting them. During our formative study, we also observed that when concentrating on their drawing, participants sometimes forgot to save steps, which they later regretted. Our prototype, therefore, prompts the child to save a step at regular intervals. These prompts are currently controlled via a wizarding interface, which allows a facilitator to issue a reminder when the child appears to be forgetting to save their steps.

After the child completes their drawing, the prototype displays an automatically generated step-based tutorial in an HTML page that the child can open, as shown in Figure 2(C). This tutorial displays the sequence of steps captured by the child and includes information on the tools used as well as any comments that the child provided while drawing (Figure 2C). Children can further modify the tutorial by editing comments, deleting unnecessary tool information, and deleting entire steps. After they finish editing the tutorial, the prototype displays the final version of the tutorial so

that the child could potentially share the tutorial with other children (e.g., friends). We leave investigating tools for sharing these captured tutorials for future research. In the Supplementary Materials, we include a short video walkthrough of the prototype.

## 6 FURTHER CONCEPT EXPLORATION & PROTOTYPE EVALUATION

Our formative study provided initial indications that children seemed open to the idea of generating tutorials for other children. In this second study, we use our higher-fidelity prototype to investigate incentives children might have to generate a tutorial for others. We were also interested in observing how they use the prototype, including how they might balance tutorial generation while focusing on their own art, how they would decompose their drawings into steps, and what type of comments they would leave for other children.

Due to the COVID-19 pandemic, we transitioned to an online study, where we interacted with participants using video conferencing software.

### 6.1 Participants

We recruited 16 participants for our study (8 girls, 8 boys), all of whom were 7-11 years old (mean age: 9.5). Due to the study being online and the numerous COVID restrictions that were in place, we recruited internationally through word-of-mouth and snowball sampling [25], beginning with the authors' personal contacts. This sampling technique resulted in participants from three different countries, including 9 participants from Canada, 6 from the US, 1 from Bangladesh. We again recruited participants with previous experience in using digital tools to draw. Participation was voluntary, and the children were informed that they could withdraw from the study any time. After receiving written consent from the parents and verbal assent from the children, we initiated the study sessions. The parents' presence on video was optional, based on the comfort level and preference of the child. In appreciation for their

participation, the family was provided with $20 in cash or as a gift card. The study was approved by our institutional research ethics board.

## 6.2 Study Tasks, Procedure, and Data Collection

To conduct the study remotely, we used video conferencing software with the parent's supervision. To enable the facilitator to act as the prototype "wizard", we used TeamViewer, which allowed the participants to access the facilitator's computer screen directly. This also meant that participants did not have to install any other software to run our prototype. Each study session was approximately 60 minutes long.

Like our initial formative study, we began by showing the child a storyboard (see the Supplementary Material) to introduce them to the idea of tutorial authoring and asked them some preliminary questions regarding their attitudes towards sharing their drawings and/or workflows. After that, the facilitator demonstrated the prototype by creating a short tutorial.

Next, we asked participants to perform the following two tasks: 1) We asked the participant to draw something of their choice. We asked them to capture their steps while drawing and told them that they could provide comments with each step if they wanted to. 2) After the child completed their drawing, we asked them to use the prototype to review the generated tutorial and make any desired modifications. We also showed the participant another tutorial of a simple drawing to get preliminary feedback on how they might feel about using others' tutorials.

After completing each task, we asked the child open-ended questions about their experience of using the prototype. We intermixed the interviews and tasks to create a more conversational atmosphere with the child as well as to provide a break from using the prototype. In piloting, we found these breaks to be particularly important with the study being online. We also asked them survey questions by adapting the Fun Toolkit survey technique [53], which has been used in previous studies with children to evaluate interface usability. Using the toolkit, we asked 10 closed, fixed-response questions, covering: i) how they felt about using the prototype's features; ii) which task they liked most, and iii) whether they would like to do each task again. The questionnaire items can be found in the Supplementary Materials. Participants completed the surveys on the facilitator's computer (using TeamViewer).

Our main source of data was obtained from the semi-structured interviews conducted throughout the study. We recorded the entire study sessions using a screen recorder to capture the interactions with the prototype. Finally, we used the surveys to elicit structured data on children's experiences with the prototype.

## 6.3 Findings: Tutorial Creation

Most of our participants (12/16) were familiar with the concept of a tutorial prior to the study and all successfully generated a step-based tutorial using the prototype. Participants' drawings included flowers, unicorns, nature scenery, ships, and favorite Lego characters. Figure 3 and Figure 4 show two example tutorials created by participants in the study (one from a 7-year-old girl and one from an 11-year-old boy).

The median number of steps generated by the participants was 7 per tutorial (min: 4; max: 10; IQR: 2). For most participants, each new element added to the drawing constituted a step. As the formation of a step was conceptual and related to elements of a child's drawing, this indicates that implementing automated step capture would be challenging. For example, simply creating a step for each tool used would have resulted in tutorials with much lower granularity than those created by our participants.

Almost all participants (15/16) provided comments with their steps. The median number of comments per tutorial was 6.5 (IQR: 3) and 14 participants provided a comment with each step.

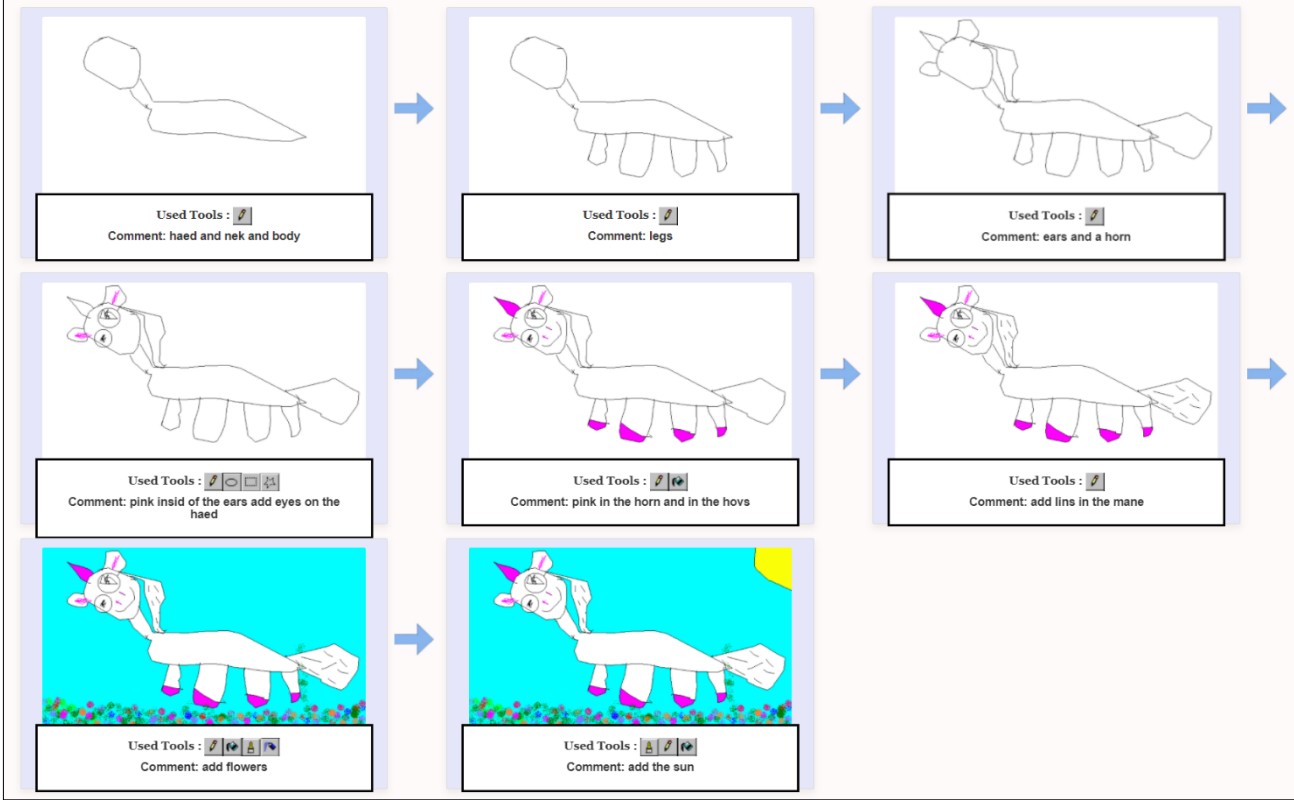

Figure 3: Tutorial authored by a 7-year-old girl (P16)

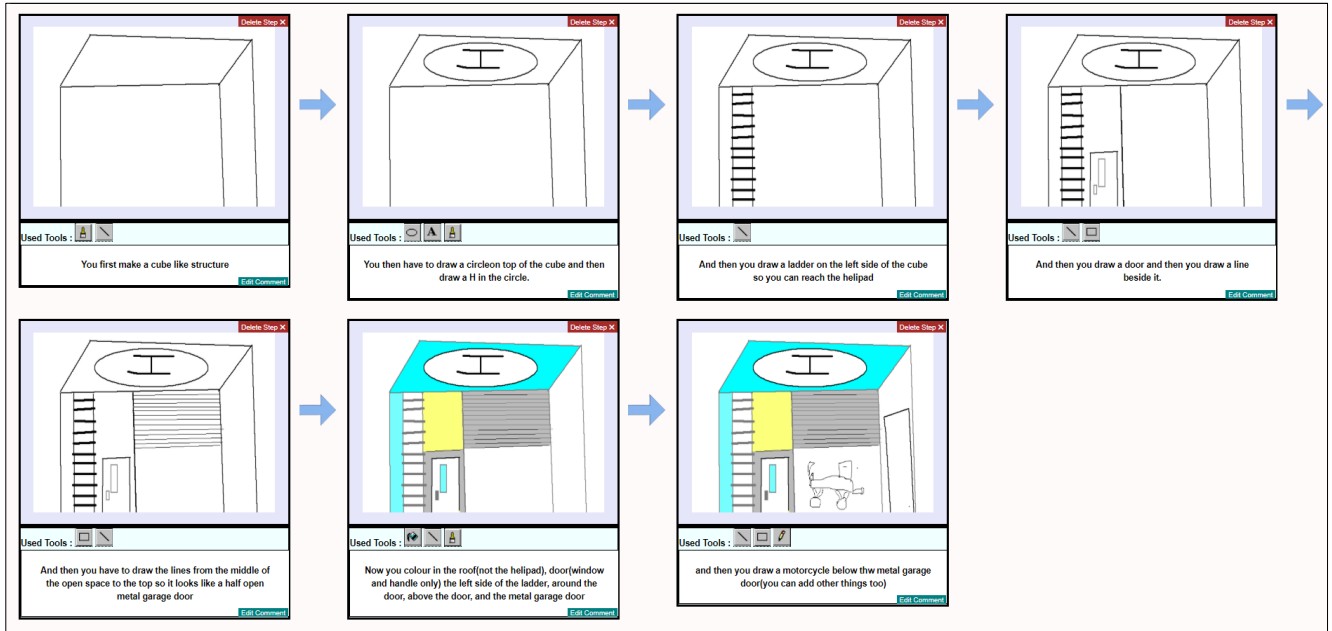

Figure 4: Tutorial authored by an 11-year-old boy (P6)

Comments often described the drawing element in the step, e.g., "the ocean", "Lego arms", "moon". Some participants provided more detailed or specific instructions with their comments, e.g., "You first make a cube-like structure", "Make a hill and colour on top", "Add texture to the grass", "Add any of your imaginary details you like". Notably, most of the comments did not focus on how to use certain features of the drawing application. Overall, we did not observe any age differences manifest themselves in the commenting style or informativeness.

Our survey findings indicated that all 16 participants felt positive about creating steps and viewing others' tutorials. 14/16 participants also felt positive about writing comments. Additionally, we found that 10/16 participants wanted to create a tutorial again; the remaining 6/16 indicated that they might be interested in doing so. However, 12/16 participants did not like to edit their tutorials once they had completed the drawing. This potentially supports our idea of capturing and generating steps when the child is creating the drawing.

During the study, we also looked for indications of how children approached the task of authoring tutorials while drawing. We observed that participants did not take time to plan out how they wanted to design a tutorial before starting to work on the drawing. Instead, they seemed to go with the flow, adding elements as they saw fit at that moment while drawing. Participants worked intently on their drawings and tutorials, suggesting that they cared about the final product. Six participants were so engrossed in working on their tutorials that we had to cut them off due to time limitations.

## 6.4 Findings: Interviews

Since we had a larger sample size and a larger volume of data, we applied a more rigorous qualitative analysis procedure than we did with our formative study, where the goal was to gather preliminary design insights. We transcribed the interview data and then analyzed it by using a bottom-up inductive approach and creating affinity diagrams to identify themes in the data [9]. While creating the affinity diagrams, one researcher initially applied open coding [15] to the quotes and then used affinity diagramming to refine the initial set of codes. This same researcher then clustered related quotes and performed axial coding [61] to identify themes. Two

Table 1: Reasons for and against creating and sharing tutorials along with the number of participants who felt this way

| Reasons for Creating and Sharing Tutorials | |
|---|---|
| Altruism | 14 participants |
| Assessing own tutorial authoring skills and seeking validation | 12 participants |
| Showcase drawing skills | 6 participants |
| To keep a record of their own drawing | 5 participants |
| **Reasons for Hesitating to Share Tutorials** | |
| Lack of confidence | 5 participants |

researchers collaboratively iterated on the raw data, clusters, and codes until clear themes emerged.

From our analysis, themes emerged related to incentives for creating and sharing, and attitudes towards child-authored tutorials, which we present below. The data collected from this study also contained insight regarding how different features of our prototype might support tutorial creation. To contextualize the quotes, we provide each participant's age and gender. As with our formative study, we report counts to give a sense of prevalence of the sentiments within our data, however, we once again acknowledge that doing so is a contentious issue within qualitative HCI research.

### 6.4.1 Perceived Incentives and Deterrents to Create and Share Tutorials

Table 1 summarizes the reasons children provided for and against the idea of creating and sharing tutorials, which addresses our second research question. All the 16 children provided at least one reason in favor, with some children providing multiple reasons in favor. Five children expressed mixed views. We elaborate on their reasons below.

*Altruism:* The main incentive to create and share tutorials for most of the participants (14/16) was altruism. There were some nuances, however, in how children expressed their desire to create tutorials as a way of helping others. For example, some of the children (4/16) wanted to help people in general by sharing their

tutorials, whereas others specifically wanted to help their friends (3/16). In terms of why they wanted to share, participants were motivated to give other kids new ideas for drawing, and felt that their tutorials could help other kids create the drawings easily:

*I'd like to show my friends so that they can get an idea of what to do next when they draw again and also, I can show them a few steps about how they can make it. ...I'd like it because it'd feel good. Like I'm helping people without even seeing them. – P2 (9-yr boy)*

Other children (4/16) liked the idea of showing kids how they might draw something differently. For these participants, it seemed to be less about showcasing the final product and more about illustrating their process.

*It's fun and lets other people learn how to draw something in another way. – P15 (10-yr girl)*

A few participants (3/16) wanted to share their tutorial only if their friends specifically asked for it. They were not confident in their drawing skills and were shy to share their drawings with others unless someone needed them.

**Assessing Own Tutorial Authoring Skills and Seeking Validation:** Some participants (7/16) wanted to share workflows with others to assess their tutorial authoring skills. If others could reproduce or make a better version of their drawings by following their tutorials, they felt that it implied that their tutorial was understandable and useful.

*I'd just wanna see how good the steps were that I made, and if they ended up making it look more realistic. – P12 (11-yr boy)*

Others (9/16) thought they would feel validated even just by having another child try their tutorial, since this would mean they produced something interesting. Knowing that others were going to view and use their tutorials to create a drawing gave them the satisfaction that their art is appreciated by others and their effort is valued.

*I'd like it because some kids like to draw, and I'd like it if they do this thing. I'd be happy too, to see that they used my tutorial. – P15 (10-yr girl)*

**Showcase Drawing Skills:** Some of the children who seemed particularly confident in their art and drawing skills, wanted to create and share their tutorials to showcase their skills (6/16). For these children, it seemed less about receiving validation and more about having an outlet to share their creativity with others.

*If I'm proud of the artwork then I'd wanna show it to other people. So that they have an opportunity to try doing art and learn. – P14 (11-yr girl)*

**To Keep a Record of Their Own Drawing:** Finally, some participants (5/16) wanted to create tutorials to keep a record for themselves so that they could review it later to recreate the drawing. It indicates that even if a child is not comfortable sharing their tutorials with others, they can still create tutorials for themselves.

*If I ever went back and reviewed it, it kinda leaves like a bookmark... Next time you can follow the steps again. – P6 (11-yr boy)*

**Lack of Confidence a Deterrent to Sharing:** Some participants (5/16) were hesitant to create and share tutorials because they believed that their drawing skills are not adequate to create tutorials, even though we did not find their drawings to be noticeably worse than the other participants. They were not confident that others would like their tutorials.

*Some of them are better at drawing and I'm scared that they're gonna judge me. – P14 (11-yr girl)*

P14 mentioned earlier in the interview that she wanted to showcase her drawing skills by sharing the artwork she is proud of. However, at the same time, she had some reservations about sharing due to her lack of confidence. This indicates that some children might be in conflict about whether to share their tutorials.

### 6.4.2 Feedback on the Design Approach

During our interviews, children provided feedback on our semi-automated tutorial authoring approach as well as on individual design elements.

**Capturing Steps Was Intuitive but Can Divert Attention:** Participants generally found saving steps while creating the drawing to be simple and intuitive (10/16). One participant mentioned that she got so accustomed to saving steps that she did it without even thinking about it.

*At one point I kinda forgot that to save step (that she's using the feature of saving steps subconsciously). I kinda got used to saving the steps. – P14 (11-yr girl)*

On the other hand, some participants (6/16) felt that saving steps distracted them from their drawing. They worried that pausing to save steps might ruin their flow and they might forget what they wanted to do.

*I was kinda in a mood. I like focusing on what I'm doing instead of stopping and doing something else. – P16 (7-yr girl)*

While an automated step capture feature could avoid this hassle, the challenge would be developing an algorithm that can detect the conceptual, element-based segmentation that children seem to want to employ when manually capturing steps.

**Mixed Reaction Towards Writing Comments:** Though all but one participant provided comments with their steps, only half of those participants (7/16) explicitly discussed the value that they saw in providing comments. They believed that comments could assist others to go through the steps and also help them remember what the steps meant if they wanted to review their own tutorials.

*Writing comments is a good way to explain it because sometimes just looking at pictures doesn't make sense. – P14 (11-yr girl)*

Some of the participants who were not as enthusiastic about commenting (4/16) found it difficult to come up with appropriate comments. They indicated that it was sometimes hard to explain the steps the way they wanted.

*Sometimes you have another way to say it in your head and it's complicated to put it in comments. – P15 (10-yr girl)*

Thus, overall, we observed mixed reactions towards commenting: some were enthusiastic about writing comments; for others, it seemed to be a source of pressure. At a minimum, this supports our decision to make commenting optional. Future versions could explore ways to assist the children who want to provide comments but struggle to verbalize their thoughts.

**Tools Are Not Always Sufficient:** The tool information provided with each of the steps was seen as useful by most participants as they felt it gave a clear idea of which tools were needed to achieve a certain effect. However, some participants wanted to provide more information regarding the tools that they used (6/16). For example, in addition to the tool name and the icon, some tools could have more details, such as brush size, the colour of the paint, etc. Future versions could explore designs that can include additional information for certain tools.

### 6.4.3 Attitudes Towards Following Other Children's Tutorials

In addition to getting insights into children's incentives to generate tutorials, we hoped to gain initial insight into how the children felt about being consumers of kid-generated tutorials. To keep sessions at a reasonable length, we showed participants a sample tutorial to elicit their opinions, but they did not have to follow a tutorial.

All children in the study responded positively to the idea of viewing another child's tutorial. The main reason for wanting to see others' tutorials was to gain new ideas and inspiration (11/16). Participants mentioned that they are sometimes unsure about what to draw, how to start, and are interested in seeing other ways to draw something. Participants also (7/16) mentioned how they can learn from others who are better at drawing by viewing their tutorials and by comparing their drawings to find potential ways to improve. One child mentioned that she wanted to make the authors feel happy that someone has tried out their tutorial. Although they were willing to view others' tutorials, three participants were not enthusiastic about the idea of following others' tutorials. They indicated that they did not like following instructions or wanted to draw something in their own way, with their own creativity.

## 6.5 Comparing Adults' vs Children's Tutorials

To get an initial sense of how an adult-authored tutorial might compare to what we saw in our study, we selected a small sample of 16 adult-authored tutorials dedicated to children from DrawingNow [72] and DragoArt [71] that fell into similar drawing categories. We picked four categories of drawings from the tutorials created by our participants – a flower, a unicorn, a robot, and a ship. We then randomly selected 8 tutorials from each of these two websites, two per category. Based on this small, curated sample, we observed both similarities and differences, which we briefly discuss here.

The median number of steps found in the selected adult-generated tutorials was 8 (min: 4; max: 14; IQR: 3), which is similar to our child-generated tutorials (median: 7). However, sometimes the adults' steps were more complicated, such that they could be decomposed into further smaller steps. On the other hand, in the majority of our child-generated tutorials, each step constituted one component of the drawing.

The tutorials on DragoArt [71] included comments with specific instructions about how the elements are added to the drawing, whereas most of the tutorials on DrawingNow [72] included only one comment with a description of the drawing and the techniques applied in the tutorial. The median number of comments per tutorial on DragoArt [71] was 7 (min: 4; max: 13; IQR: 3), similar to our participants (median: 6.5). As expected, the comments in the adult-generated tutorials were often more detailed than the child-created ones, however, a few of our participants did craft detailed comments (e.g., see Figure 4). Like our child participants, the adult authors did not focus on the features or tools of the drawing application in their comments.

The most striking difference we observed was that the adult-authored tutorials mostly followed a structured way of drawing, starting with a workable frame to make the drawing process easier. More than half of the selected adult-authored tutorials showed this pattern. On the other hand, our participants took a less structured approach, allowing their drawings to move in creative directions. One potential reason for this difference might be the fact that children's main goal was to create a drawing from their imagination and share the process with others, as opposed to teaching specific drawing techniques.

## 7 DISCUSSION, LIMITATIONS, AND FUTURE WORK

Findings from our second study suggest that most of the children were interested in and capable of authoring drawing tutorials. The study findings also shed light on children's perceived incentives to author and ultimately share their tutorials, which included helping their peers and other social incentives (e.g., seeking validation and showcasing skills). Some also wanted to maintain a record for their own purposes. We were surprised by the extent that their motivations mirrored those found in prior work on adult populations. For example, altruism is an intrinsic motivator for adults who share their knowledge online [65]. Similar to the incentive of 'showcase drawing skills', adults also author tutorials to showcase the workflows they find interesting [50]. Self-efficacy is another important consideration [66]. In our studies, we noticed that children's level of confidence in their drawing abilities seemed to affect their attitudes towards sharing.

Our findings indicate that a semi-automated tutorial authoring system can potentially enable children to generate step-based tutorials. In terms of important design considerations, most children in our study responded positively to the idea of creating a tutorial while they were drawing. Further, they found the post-hoc modifications to be the least fun activity of the study session. This suggests that interleaving tutorial generation with the principal activity is a promising design direction. We saw that children wanted to control the granularity of their steps, but sometimes became so engrossed in the drawing activity that they forgot to do so. Adaptive prompts or automated step-capture features could potentially address this but would need to consider the characteristics and tendencies of the child artist. Our findings also suggest that children appreciated the ability to annotate their steps, however, some found it difficult to craft good comments. Future work could therefore consider ways to scaffold this process, for example, through sample comments or comment templates.

Children seemed interested and open to the idea of using another child's tutorial, however, further study is needed to understand the relative advantages and disadvantages of the child- vs. adult-authored tutorials for this type of creative activity. When analyzing the tutorials that children produced and comparing them to a small sample of adult-authored tutorials, we observed that adult authors tended to put more effort into structuring the drawing process and crafting comments, whereas children seemed to focus on their drawings and generated the tutorial as a by-product of that activity. However, this might be an artifact of our study design, which did not involve a dedicated tutorial planning phase. The child-created tutorials also involved more straightforward drawings and simpler comments than the adult-created ones, which might be easier for younger children to follow. Future research should investigate these differences in a more structured and systematic way as well as how children experience the tutorials. For example, it is possible that adult-authored tutorials are better at teaching drawing skills and specific techniques, whereas children's tutorials might be more relatable and inspire creativity.

We conducted our second study online due to the COVID-19 restrictions, which introduced some limitations. For example, participants were sometimes distracted by siblings, some experienced internet issues, and some parents had difficulties setting up the study. A recent study investigating online synchronous co-design with children during the pandemic also identified these factors as impacting children's interaction during study sessions [39]. While designing the online study, we had to be particularly mindful of study session length due to the video conferencing fatigue. For example, we had originally intended to have children try a previously created tutorial to elicit grounded data on their perceptions, a task that we eliminated after pilot tests.

Despite the difficulties, we found that participants in our studies were as or even more engaged in the interviews than they were in our initial lab-based study. We suspect that being in the familiar environment of their home helped make the children comfortable in expressing their thoughts.

Given that our second study was conducted online, we were able to recruit internationally from three different countries, which introduced diversity into our participant pool. However, diverse backgrounds can impact interview responses [22] and could potentially have made our investigation less focused than it would have been with a more locally recruited population. While we did not see any noteworthy differences in how participants from the three different countries approached tutorial authoring and sharing, future work should investigate the generalizability of our findings to a larger sample of children with both similar and different backgrounds.

While building a child-centric sharing platform is beyond the scope of this work, the overlap in our participants' motivations for sharing their tutorials with prior results on adult tutorial sharing, suggests opportunities to learn from prior adult-centric research on how to motivate sharing online. For example, positive voting and textual comments have been shown to encourage adults to contribute [13]. Future work can explore the extent to which these prior approaches could also encourage a range of children to share their digital art workflows online, or conversely, if new child-centric approaches are needed. In the future, a longitudinal study could enable us to investigate how the act of sharing one's art tutorials impacts a child's sense of self-accomplishment. Additionally, such a study could reveal interesting insights into how children might improve their drawing skills through a combination of teaching others, viewing other workflows, and reflecting on their own process.

Future work could also explore alternative uses of this type of drawing-capture approach. For example, one child in our study proposed the idea of using the system to create an illustrated story with her friends. In addition to acting as a creative outlet, prior work in the domain of programming found that storytelling helped children learn the concepts [36]. Finally, it would be interesting to explore the generalizability of our approach to other creative activities that involve complex software, such as 3D modelling for child-oriented makerspaces.

## 8 CONCLUSION

In this paper, we present the participatory design and evaluation of a children's tutorial authoring system for digital art. Findings from our studies illustrate the potential for children to be engaged and motivated by this form of peer-based help and knowledge sharing, with potential applications to other domains (e.g., helping children create programming tutorials). Our approach is also but one way to provide children with tools to share aspects of their creative process with others. Future work should explore new ways for children to communicate their digital art ideas and skills with their peers and connect with other children in positive online communities. Future work should also study the role of such communities in fostering important social skills.

### ACKNOWLEDGEMENTS

We would like to thank Joanna McGrenere and Parmit K. Chilana for their feedback on early phases of this research. We would also like to thank the Natural Sciences and Engineering Research Council of Canada (NSERC) for funding this work.

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
