# OpenReview forum: "Tutorials for Children by Children: Design and Evaluation of a Children’s Tutorial Authoring Tool for Digital Art"
_graphicsinterface.org/Graphics_Interface/2022/Conference — GI 2022_

### Official Review · Reviewer_Jz2e · 2022-04-13
**This is a nice exploration of how children approach explaining their creative process to others.**

**Rating:** 9
**Confidence:** 4

**Review:**

This multi-stage exploration is well thought-out and conducted. Any of the critiques that I have are minor. I think this paper provides a nice example of how to explore the practices and preferences of a vulnerable and underserved population. I recommend acceptance.

The research designs were thoughtfully constructed and capture multiple stakeholders' perspectives.

Potential Improvements
- While the targeted age range is justified, the breadth of developmental stages it covers raises questions and should be acknowledged in a limitations section.
 - More discussion could be given to the choice to recruit children who make art rather than those who also make tutorials or otherwise share their process since that is the goal of the developed system
- There is a conflation between culture and nationality. I recommend only discussing nationality if there isn't the space to discuss culture in a more nuanced way. Also, please make the nationality information of participants from the first study known so that there is parallelism in the reporting of participant background.
- When looking for indications of how children approached the task, what was considered; was there a theoretical lens; and how were their approaches determined?
- It would be nice to see the information that is presented in table 1 on a per-participant level. Perhaps a visualization that resembles a heat map or a table with checkmarks would work.
- I would like to see more discussion of the role of self-efficacy. I would also like to see it better integrated into the discussion. However, I'm not sure that there's room for this

---

### Official Review · Reviewer_aq2f · 2022-04-13
**Moderate revisions improved this paper such that I can lean towards acceptance**

**Rating:** 6
**Confidence:** 3

**Review:**

I am reviewing a revision of this paper and mainly cover my two prior concerns. Overall, there are a number of changes, which while not substantial, do address some of the raised concerns. Overall they strengthen the paper a bit towards a soft accept.

1) Insights about tool design process.

Changes:
- The authors have added some details to section 3, indicating where the design of the tool comes from some additional rationale, but not giving in-depth rationale for features that might be necessary for publication.
- Fig 1 is improved with more descriptive annotations and a more detailed caption.
- Section 5 has some additional details about prototype functionality (reminding users to save) - this is the kind of logic I think needs to be throughout this paper.


2) Qualitative analysis - methods and reporting.
Changes:
 - The authors have added details about their open coding process, but no specific methodology in study 1. This is enough to describe what the authors did, but will rely on the rigor of reporting in the results section. Good that the authors specify their position on coding counts in their results.
- In study 2, more substantial methods are used.

---

### Official Review · Reviewer_S37s · 2022-04-13
**Insightful and Refreshing Paper, with prior shortcomings addressed**

**Rating:** 9
**Confidence:** 3

**Review:**

I have previously reviewed this paper, and I remain very positive about both the structure and insight presented here. This is a well-written and well-organized paper on an overlooked topic in graphics, i.e. designing digital tools to support children's creativity. The authors present substantial related research, which is itself an important contribution, as it helps raise awareness of related topics in the graphics community. Further, the authors carefully design two studies and present detailed and insightful findings that can foster future research. The idea itself, of allowing children to author tutorials, is brilliant, and the findings of this study, such as children's eagerness to participate and their reasons for doing so, can encourage research and product design in this area. In additions, interesting findings are summarized and substantiated very well, bringing value to anyone trying to design such a tool in practice. It is a refreshing and interesting read.

The main criticisms of the previous reviews, i.e. insufficiently well-described interface, has been addressed with a better figure, and I believe sufficient details are now provided. Other considerations, such as international participant recruitment and additional references have also been addressed.

Minor:
- I do not think the paper is better served by putting a previously separate section on comparisons with adult tutorials into the discussion (following previous critique). Even if not a major part of the contribution, this analysis can inspire future research directions, and I would keep it as a separate small sub-section. If not a part of the major contribution, details can go into the appendix.
- On a related note, currently section 7 has too many things. Details about participant recruitment should probably not be here, unless directly relevant to limitations. Please rework this part to neatly summarize main findings.

Beyond the scope of this work, it would be really interesting to see future work explore if authoring tutorials can also help children gain insight into their own process and improve their skills by teaching others, as is often true for adults. I hope this paper will invite more work into this area, as design of engaging digital experiences for young minds is deeply needed in our screen-heavy world.

---

### Decision · Program_Chairs · 2022-04-17

Accept